# Store-Operated Ca^2+^ Entry in Skeletal Muscle Contributes to the Increase in Body Temperature during Exertional Stress

**DOI:** 10.3390/ijms23073772

**Published:** 2022-03-29

**Authors:** Barbara Girolami, Matteo Serano, Antonio Michelucci, Laura Pietrangelo, Feliciano Protasi

**Affiliations:** CAST, Center for Advanced Studies and Technology & DMSI, Department of Medicine and Aging Sciences, University G. d’Annunzio of Chieti-Pescara, 66100 Chieti, Italy; barbara.girolami@unich.it (B.G.); matteo.serano@unich.it (M.S.); antonio.michelucci@unich.it (A.M.); laura.pietrangelo@unich.it (L.P.)

**Keywords:** Ca^2+^ entry unit (CEU), heat stroke (HS), malignant hyperthermia (MH), store-operated Ca^2+^ entry (SOCE)

## Abstract

Exertional heat stroke (HS) is a hyperthermic crisis triggered by an excessive accumulation of Ca^2+^ in skeletal muscle fibers. We demonstrated that exercise leads to the formation of calcium entry units (CEUs), which are intracellular junctions that reduce muscle fatigue by promoting the recovery of extracellular Ca^2+^ via store-operated Ca^2+^ entry (SOCE). Here, we tested the hypothesis that exercise-induced assembly of CEUs may increase the risk of HS when physical activity is performed in adverse environmental conditions (high temperature and humidity). Adult mice were: (a) first, divided into three experimental groups: control, trained-1 month (voluntary running in wheel cages), and acutely exercised-1 h (incremental treadmill run); and (b) then subjected to an exertional stress (ES) protocol, a treadmill run in an environmental chamber at 34 °C and 40% humidity. The internal temperature of the mice at the end of the ES was higher in both pre-exercised groups. During an ES ex-vivo protocol, extensor digitorum longus(EDL) muscles from the trained-1 month and exercised-1 h mice generated greater basal tension than in the control and were those that contained a greater number of CEUs, assessed by electron microscopy. The data collected suggest that the entry of Ca^2+^ from extracellular space via CEUs could contribute to exertional HS when exercise is performed in adverse environmental conditions.

## 1. Introduction

Exertional heat stroke (HS) is a hyperthermic crisis triggered by strenuous exercise [1,2] that shares common features with overheating episodes triggered by exposure to elevated environmental temperatures (crises known as environmental HS) or with hyperthermic episodes triggered by febrile illness [3]. Exertional HS has also been related to malignant hyperthermia (MH) susceptibility, which causes life-threatening hypermetabolic responses following the administration of halogenated/volatile anesthetics, such as halothane or isoflurane [4,5,6]. Individuals are diagnosed as MH susceptible if they have experienced hyperthermic crises during anesthesia [7,8] and score positive on a diagnostic in-vitro contracture test (IVCT) [9,10]. It is widely accepted that MH and exertional HS crises are caused by an excessive elevation of Ca^2+^ levels in skeletal muscle fibers [11,12,13,14]. Most, but not all, families affected by MH susceptibility have been associated with mutations in the ryanodine receptor type-1 gene (RYR1), which encodes for the sarcoplasmic reticulum (SR) Ca^2+^ release channel of skeletal muscle [15,16]. The ryanodine receptor type-1 (RyR1) is a key protein in excitation–contraction (EC) coupling, the process that controls release of Ca^2+^ from the SR during muscle activation [17,18].

The correlation between MH and exertional HS is strongly supported by studies in animals. In porcine stress syndrome (PSS), pigs carrying a point mutation in the RYR1 gene trigger MH episodes in response to halothane or during exposure to elevated environmental temperature and/or emotional/physical stress [19,20]. Moreover, knock-in mice that carry gain-of-function mutations in the RYR1 gene that are causative of MH in humans (R163C and Y522S), and knockout mice that lack Calsequestrin-1 [21] are susceptible to lethal overheating crises when exposed to halogenated anesthetics, elevated temperature, and exertional stress [22,23,24,25,26,27].

The levels of intracellular Ca^2+^ in skeletal muscle fibers, though, are not regulated just by EC coupling; for example, store-operated Ca^2+^ entry (SOCE) also significantly contributes to the regulation of intracellular Ca^2+^ homeostasis. Data in the literature suggest a possible involvement of SOCE in MH episodes: (i) SOCE is upregulated in human skeletal fibers isolated from MH susceptible patients [28]; (ii) SOCE is enhanced in mouse animal models of MH and accelerated by temperature [29]; and (iii) the standardized procedure to determine MH susceptibility (i.e., IVCT) is performed in a solution containing external Ca^2+^ (and requires the presence of Ca^2+^ to discriminate between positive and negative outcomes) [30]. However, whether SOCE contributes to exertional HS still remains to be determined.

SOCE is a ubiquitous mechanism triggered by the depletion of intracellular Ca^2+^ stores (endoplasmic/sarcoplasmic reticulum, ER and SR) [31,32] and was measured for the first time in skeletal muscle about 20 years ago [33,34,35]. SOCE in muscle, as in other cells, is mainly mediated by two proteins: (a) stromal interaction molecule-1 (STIM1), a protein placed in the ER/SR membrane, which has intra-luminal domain that acts as Ca^2+^ sensor [36,37,38]; and (b) Orai1, a Ca^2+^ release-activated Ca^2+^ (CRAC) channel placed in external membranes or transverse-tubules (TT) [39,40]. It is now generally accepted that SOCE is especially important for muscle function during repetitive stimulation, as recovery of external Ca^2+^ is crucial to limit muscle fatigue [41,42,43,44,45,46].

The sites of STIM1/Orai1 interaction in skeletal muscle fibers has been long debated, as initially it was proposed that SOCE would occur in triads, the intracellular junctions deputed to EC coupling [17]. Though, this initial hypothesis was not confirmed by recent experimental evidence collected in our laboratory between 2017 and 2020 [44,47,48,49,50,51,52]. New intracellular junctions named Ca^2+^ entry units (CEUs), formed by the association of TT and SR at the I band, were identified as the sites of interaction between STIM1 and Orai1 during SOCE. CEUs were first shown to promote increased STIM1 and Orai1 colocalization and to increase fatigue resistance in presence of extracellular Ca^2+^ [44]. Then, the final evidence of CEUs as the sites of SOCE came with the demonstration that the presence of these junctions was associated with an increased rate on Mn^2+^ quench, the gold standard technique used to assess entry of divalent cations from the extracellular space [49,50]. Finally, as proof of principle, CEUs are constitutively assembled in the muscle fibers of CASQ1-null mice [50], which quickly undergo SR depletion (the putative trigger of STIM1 aggregation and SOCE activation) during repetitive stimulation [53].

To determine whether SOCE may play a role in exertional HS [28,29,30], in the present work, we tested the hypothesis that Ca^2+^ entry via CEUs may increase the risk of hyperthermic crisis when physical activity is performed in challenging environmental conditions. The results collected in this study show for the first time how exercise-induced assembly of CEUs results in: (a) hypersensitivity of muscle to develop contracture and (b) excessive increase in the internal temperature of mice subjected to exertional stress (ES).

## 2. Results

*The Body Temperature of Trained and Pre-Exercised Mice Increases More than in Control Mice during ES.* The three groups of animals included in the study (control, trained-1 m, and exercised-1 h) were subjected to an ES protocol based on incremental running [27]. During the ES protocol, the treadmill was placed in a climatic chamber in which the temperature and humidity were maintained at constant values of 34 °C and 40%, respectively (Appendix A). Measurements of body temperature were performed at two different time points: the beginning and the end of the ES protocol (respectively, T_0_ and T_f_), using either a rectal thermometer (not shown) or a commercially available infrared thermometer (Appendix A). The second system was employed with the goal of testing and validating a less invasive way to measure the internal body temperature of mice in future studies. The results obtained with the two systems were comparable (compare Figure 1, Appendix A), not for the absolute temperature recorded (the skin temperature was on average 2.4 degree lower), but for the change in body temperature during the ES and for the differences between the different groups of animals included in the study.

The results collected indicate that:The increase in body temperature was greater in the trained-1 m and exercised-1 h mice than in the controls. The difference with the control mice was statistically significant, both as an absolute temperature (Figure 1A: 38.9 ± 0.2 in trained-1 m and 38.8 ± 0.5 in exercised-1 h vs. 37.9 ± 0.2 in control) and as ΔT, i.e., the difference between the temperatures measured at the beginning (T_0_) and at the end (T_f_) of the experiment (Figure 1B: 2.7 ± 0.2 in trained-1 m and 2.3 ± 0.1 in exercised-1 h vs. 1.7 ± 0.1 in control). See also Appendix A. Results collected using the infrared thermometer display a similar trend, both as an absolute temperature (Appendix A: 36.5 ± 0.4 in trained-1 m and 36.3 ± 0.5 in exercised-1 h vs. 35.6 ± 0.3 in control) and as ΔT (Appendix A: 2.7 ± 0.3 in trained-1 m and 2.3 ± 0.1 in exercised-1 h vs. 1.6 ± 0.3 in control).Note that all graphs (Figure 1 and Appendix A) indicate an increase in body temperature that is slightly greater in the trained-1 m mice than in the exercised-1 h mice; this difference, though, was not statistically significant.

*Extensor digitorum longus (EDL) muscles dissected from trained and pre-exercised mice develop a greater basal tension than those from the control during an ex-vivo ES protocol.* We assessed the development of contractures in response to a high-frequency tetanic stimulation protocol (150 consecutive pulses lasting 500 ms each at 80 Hz, delivered every 10 s; duty cycle 0.05) in intact EDL muscles isolated from control, trained-1 m, and exercised-1 h mice. When exposed to this protocol (30 min of total duration at 30 °C), the EDL muscles from all three groups displayed an increment of basal tension. However, significant differences were observed in either the kinetic or the amplitude of contractures (Figure 2A,E). Indeed, while the muscles from the control mice showed an increase in basal tension that started approximately after 100 tetani from the beginning of the experiment, the EDL muscles from both the trained and acutely exercised animals exhibited a rise in basal tension that started significantly earlier compared to the controls (approximately after 60–70 tetani from the beginning of the experiment), resulting in the development of a full contracture in the second half of the stimulation protocol (Figure 2A). Specifically, at the 150th tetanus, the relative basal tension of the EDL muscles from trained-1 m and exercised-1 h mice was increased ~2.5 times (vs. ~1.7 times in control) (Figure 2B).

To assess the relative contribution of extracellular Ca^2+^ to the development of tension shown in Figure 2A, the experiments were then repeated in the presence of 10 µM BTP-2, an established inhibitor of SOCE [44,49,50,54]. The results collected in these experimental conditions showed how the rise in basal tension decreased in all three groups of samples; it decreased completely only in control (Figure 2B–D,F–H).

In parallel experiments (Appendix A), we also exposed intact EDL muscles from control, trained-1 m, and exercised-1 h mice to an ex-vivo heat stress protocol to verify if temperature alone could cause the development of contractures (Appendix A). In this type of experiment, intact EDL muscles were subjected to steps of temperature (2 °C each step every 5 min from 30 to 42 °C), while being stimulated at a low frequency (1 Hz) in order to generate single twitches. Only the EDL muscles from the exercised-1 h mice showed a significant increase in basal tension, starting at 38 °C, with a subsequent linear increase that reached a value of approximately 1.77 ± 0.1 times higher than the initial tension at 42 °C (Appendix A). Alternatively, the EDL muscles excised from the control and trained-1 m mice generated a smaller increase in basal tension (Appendix A). Note that the EDL muscles from the trained-1 m mice exhibited a time course of basal tension identical to that of control muscles (Appendix A). Supplementation with 10 µM BTP-2 reduced the development of tension in all three groups of samples; it reduced completely in the control and trained-1 m mice (Appendix A). The outcome of these experiments will be examined in detail in the Discussion section.

*EDL fibers of trained and pre-exercised mice contain more CEUs.* We have recently demonstrated that SOCE is mediated by CEUs, structures that are formed at the I band by two distinct components: SR-stacks and I band extensions of TTs [44,47]. Here, we quantified SR-stacks and extensions of TTs before and after the ES protocol in all three experimental groups (Figure 3 and Appendix A), after checking in histological longitudinal sections of EDL fibers, that no structural damage occurred in basal condition and after the ES protocol in all three animal groups (Appendix A).

Quantitative analysis indicates that, in control conditions, SR-stacks (Figure 3A) are seen in few fibers (Figure 3D) and, where present, their number/area is low (Figure 3E). Regarding the size, the SR-stacks of control fibers are shorter that those formed during training or exercise (Figure 3F) and, on average, are formed by two elements (Figure 3G). Alternatively, in the muscles of mice trained-1 m or exercised-1 h, SR-stacks (Figure 3B,C) were more frequent: SR-stacks are present in a larger percentage of fibers (Figure 3D), and, where present, their number/area was about four times greater than in the controls (Figure 3E). Regarding their size, following training or acute exercise, the SR-stacks become longer (Figure 3F) and are more often formed by three elements (Figure 3G).

Note that while assembly of SR-stacks during acute exercise was already reported [44,49], the assembly of SR-stacks following voluntary training in wheel cages is novel, i.e., not previously reported (Figure 3B). To quantify the presence of TTs at the I band, we stained muscle fibers with ferrocyanide, which creates a dark precipitate inside their lumen (Figure 4).

In fibers from the control mice, staining of TTs is mostly confined in triads (Figure 4A and inset), and quantitative analysis confirms that TT extensions at the I band (and their contact with SR-stacks) is limited (Figure 4D,E). Nevertheless, following training and acute exercise, stained TTs were visually more frequent (Figure 4B,C); an observation confirmed by the quantitative analysis (Figure 4D,E).

We also performed the quantitative analysis of SR-stacks and I band extension of TT network after the ES protocol (Appendix A). In general, the ES protocol increased most parameters (number of fibers with SR-stacks, number of stacks/area, TT I band extensions, and, finally, the contacts between TT and SR; Appendix A), with the exception of the size of the SR-stacks, which remained approximately the same (Appendix A). This quantitative analysis revealed that the difference between the control and trained-1 m/exercised-1 h that existed before the ES protocol (Figure 3 and Figure 4) is diminished when the analysis if performed post-ES protocol. This could be explained by the fact that the control mice, when exposed to the ES protocol (which is itself an incremental running protocol), started to assemble CEUs while exercising.

*Number and volume of mitochondria is similar in control and exercised-1 h mice but significantly higher in mice trained-1 m.* Cellular respiration, which increases significantly in muscle fibers during exercise, takes place in the mitochondria and is an exothermic reaction that releases heat [55,56]. As during ES protocol, there is a significant increase in the body temperature of mice (Figure 1 and Appendix A). Here, we performed a quantitative analysis of the mitochondria (Figure 5) by measuring in EM the number of the mitochondria/area and the relative fiber volume occupied by these organelles. A similar method of analysis was used in a previous publication, both in the muscle of mice and in human biopsies [57,58,59]. Data in Figure 5 and Appendix A indicate that both values are significantly higher in the mice trained-1 m, while they are similar in the control and exercised-1 h mice. In detail: (a) numberof mitochondria/100 μm^2^: control 26.8 ± 0.8, trained-1 m 41.1 ± 1.2, exercised-1 h 29.2 ± 0.9; (b) relative mitochondrial volume (expressed as % of total fiber volume): control 3.8 ± 0.1, trained-1 m 7.1 ± 0.2, exercised-1 h 4.0 ± 0.1. These results are not surprising, since it is well known that regular aerobic training promotes mitochondriogenesis, which increases mitochondrial number and volume [60,61,62,63].

## 3. Discussion

*State of the Art.* Exertional heat stroke (HS) is a life-threatening overheating episode, which affects individuals while performing intense physical exercise in hostile environmental conditions (i.e., high temperatures often combined with excessive humidity) [1,2]. Experiments carried out in mice demonstrate that exertional HS share common features with hyperthermic episodes triggered by the administration of volatile anesthetics (MH crisis) or exposure to heat (crises known as environmental HS) [4,5,6]. Indeed, we demonstrated that, in mice genetically predisposed to MH crises and environmental HS, overheating episodes are also triggered by exertional stress [27]. These episodes share common molecular mechanisms with anesthetic-induced MH crises: an excessive leak of Ca^2+^ from the SR and overgeneration of reactive species of oxygen and nitrogen (ROS and NOS) [24,27,64]. However, some issues remain unresolved: (a) what is the role of extracellular Ca^2+^ in the overgeneration of tension and heat by muscle? (b) Are there factors that cause predisposition to exertional HS, independent from the genetic defects causing predisposition to MH susceptibility?

The fact that Ca^2+^ influx from the extracellular space could play a role in the complex events that lead to MH crises is suggested by the literature. As detailed in the introduction: (i) SOCE is enhanced in human skeletal fibers from patients with MH [28] and in fibers from both RYR1^Y522S/WT^ and CASQ1-null mice [29]; (ii) the standardized procedure to determine MH susceptibility (i.e., IVCT) is performed in a solution containing external Ca^2+^ (and requires the presence of Ca^2+^ to discriminate between positive and negative outcomes) [19,30,65]. However, whether SOCE contributes to exertional HS has never been addressed before.

As we have recently discovered that exercise induces remodeling of SR and TT (to form CEUs), increased STIM1/Orai1 aggregation, enhanced Ca^2+^ entry via SOCE, and, finally, improved fatigue resistance [44,47,48,49,50], in the present work, we tested for the first time the hypothesis that exercised-induced assembly of CEUs may contribute to excessive generation of heat in mice subjected to ES in hot/humid environmental conditions. To accomplish this, we compared mice and isolated muscles from control animals and from animals with pre-assembled CEUs (a. trained for 1 month in wheel cages or b. exercised for 1 h with a protocol that we know induces CEU assembly).

*Main finding of the study.* The results collected show how mice trained-1 m or exercised-1 h displayed a greater increase in body temperature than controls during the ES protocol (Figure 1, Appendix A). In addition, their EDL muscles (which contain more of the elements that form CEUs, i.e., SR-stacks and TT extensions at the I band), before the beginning of the ES protocol (Figure 3 and Figure 4), developed more tension resulting in contracture during an ex-vivo ES protocol (Figure 2). Importantly, the generation of passive tension in muscle containing a greater number of pre-assembled CEUs was significantly reduced by the presence of a SOCE blocker, i.e., BTP-2. The ability of BTP-2 to reduce the development of contracture (Figure 2) speaks in favor of a significant contribution of extracellular Ca^2+^ in the generation of basal tension during ES. Note that some entry of extracellular Ca^2+^ is also present in EDL muscles from control mice, as the passive tension during the ex-vivo ES protocol is also reduced by BTP-2 (Figure 2 and Appendix A); this is consistent with previous findings, which show how CEUs are also present in the muscle of control animals (even if smaller and in a reduced number) [44].

*Additional findings*. The results collected in this study supported collection of two other complementary results, which deserve to be discussed:(a)the results collected in Appendix A indicate that measuring the temperature of animals using a simple infrared thermometer for cutaneous temperature (Appendix A) can be a valid and less invasive alternative method for avoiding the use of the rectal thermometer in future studies.(b)The assembly of CEUs is also promoted by voluntary training in wheel cages; this result is novel, i.e., it was not previously reported (Figure 3 and Figure 4). The following remain to be determined: (i) whether CEUs assembled by training are more stable that those formed during acute exercise, and (ii) whether training promotes an increased expression of STIM1 and Orai1. Hence, the effect of prolonged training will need to be investigated more in depth. Though, the fact that, during heat stress (Appendix A), EDL from trained-1 m mice did not behave differently from those of the controls suggests that TTs do retract quickly following exercise in the trained-1 m mice as well (see next section for additional detail).

*Differences between acutely pre-exercised mice and mice trained in wheel cages.* There are two main differences between the outcomes of experiments in exercised-1 h and trained-1 m mice that are worth discussing:the mice trained-1 m in wheel cages, when exposed to the ES protocol, reached a final temperature that is slightly higher than that of the mice exercised-1 h (Figure 1). This result is consistent both when measuring the internal temperature with a rectal thermometer and the cutaneous temperature with the infrared thermometer (compare Figure 1 and Appendix A). To determine a possible explanation for this outcome, additional investigation is needed. However, one of the possible explanations is the fact that the muscle in the mice trained-1 m has a higher number/volume of mitochondria than in the other two groups (Figure 5). Indeed, it is well known that training increases mitochondrial volume and number [60,61,62,63,64] and that the augmented mitochondrial activity during aerobic respiration could generate additional heat [55,56].In the ex-vivo heat-stress protocol (Appendix A), we exposed EDL muscles to increasing steps of temperature to assess whether the temperature alone was a relevant factor in determining hyper-contracture of muscle. The results indicate that the EDL muscles dissected from mice exercised-1 h (by a treadmill run protocol which promotes the assembly of the CEUs; [44]) developed a greater basal tension compared to that of the controls, while the EDL muscles from trained-1 m mice did not behave differently from those of the controls. The explanation for this unexpected result could reside in the following observation: the extension of TTs at the I band is more pronounced in the mice exercised-1 h than in the mice trained-1 m (Figure 4). The reason for this could be the plasticity of TTs, which allows their retraction from the I band following exercise [49]. This considered, (A) EDL muscles from exercised-1 h mice were dissected for functional studies and EM immediately after the protocol of incremental running designed to induce fatigue and promote assembly of CEUs (as in [44,49]); in this case, the TTs did not have much time to retract before being dissected for experiments. (B) Alternatively, the muscles of mice trained-1 m were dissected from mice kept in cages equipped with running wheels and likely did not exercise intensively immediately before the collection of muscle samples for functional studies and EM. In the latter case, the TTs had enough time to retract, as shown by quantitative analysis in Figure 4. As the assembly of functional CEUs requires the presence of TTs at the I band and of their contact with SR-stacks [49], the EDL muscles dissected from mice trained-1 m are less predisposed (than muscles from mice exercised-1 h) to develop contractures solely from being exposed to heat (Appendix A).

## 4. Materials and Methods

### 4.1. Animals

All procedures and experiments in this study were conducted according to the National Committee for the Protection of Animals Used for Scientific Purposes (D. lgs n.26/2014) and were approved by the Italian Ministry of Health (1202-2020/PR). Animals were housed in microisolator cages at 20 °C in a 12-h light/dark cycle and provided free access to standard chow and water. Wild type (WT) C57bl/6 male mice were randomly assigned to the following three experimental groups:

*(a) Control mice,* i.e., WT adult mice (4 months of age);

*(b) Trained mice (trained-1 m):* WT mice were housed in wheel cages for voluntary running for 1 month, from 3 to 4 months of age. During the training period, animals were individually accommodated in a 16-station home cage running wheel (Columbus Instruments, Columbus, OH, USA), and had free access to standard chow and water. Each individual wheel cage was connected to a computer through a magnetic indicator and a sensor. Running activity of mice was monitored via CMI software (Columbus Instruments, Columbus, OH, USA).

*(c) Pre-exercised mice (exercised-1 h):* Four-month-old WT animals were subjected to a single bout of incremental running protocol at room temperature (RT) on a treadmill (Columbus Instruments, Columbus, OH, USA) with 0° incline. This protocol is known to induce assembly of CEUs [44]. Briefly, immediately after 10 min of warm-up at the speed of 5 m/min, mice were subjected to an exercise protocol consisting of an initial running for 25 min at the speed of 10 m/min, followed by 20 min at 15 m/min, 15 min at 20 m/min, and then, in the last five minutes of running, the speed was increased of an additional 1 m/min at 1 min intervals, until reaching the final speed of 25 m/min.

All animals were sacrificed by cervical dislocation at four months of age, as approved by the D. lgs. n.26/2014.

### 4.2. In Vivo-Experiments

*Exertional stress (ES) protocol.* Control, trained-1 m, and exercised-1 h mice were subjected to an ES protocol (beginning no later than 15 min from CEUs-assembly protocol) carried out on a treadmill (Columbus Instruments, Columbus, OH, USA) placed in a climatic chamber in which temperature and humidity were maintained at constant values of 34 °C and 40%, respectively. See [27] for additional detail.

Briefly, during the ES, the protocol speed of running on the treadmill was initially set at 5 m/min for 5 min and then increased as follows: 10 m/min for 10 min, 15 m/min for another 10 min, and then 20 m/min for 10 min. Finally, the speed was increased an additional 1 m/min at 1 min intervals to reach the final speed of 30 m/min.

A mild electrical stimulus (0.5 mA) was applied to mice that stepped off the treadmill to keep them exercising. When an animal was markedly unable to maintain running speed and showed signs of dyspnea, its running was stopped, and the animal was euthanized.

*Core temperature recordings.* Core body temperature in the control, trained-1 m, and exercised-1 h mice was measured at two different time points of the ES protocol (at the beginning and at the end, respectively, T_0_ and T_f_). T_0_ and T_f_ core temperatures were recorded using a rectal thermometer (4-channel thermometer TM-946; XS instruments, Modena, Italy).

In order to test a less invasive method and refine the technique, the body temperature was also recorded using an infrared thermometer (CW364, RS Components, Milan, Italy). Briefly, cutaneous temperature was measured at T_0_ and T_f_ by placing the thermometer at a distance of about 15 cm from the animal.

### 4.3. In-Vitro Contracture Test (IVCT)

EDL muscles from control, trained-1 m, and exercised-1 h mice were subjected to an in-vitro contracture test (IVCT) with two different protocols of ex-vivo stimulation using the Aurora Muscle Physiology System (1200A: Isolated Muscle System, Aurora Scientific, Aurora, ON, Canada). EDL muscles were attached to a servo motor and force transducer (model 1200A, Aurora Scientific) and electrically stimulated using two platinum electrodes in a chamber continuously perfused with oxygenated Ringer’s solution containing (in mM): 137 NaCl, 5 KCl, 1.2 NaH_2_PO_4_, 1 MgSO_4_, 2 CaCl_2_,10 glucose, and 24 NaHCO_3_ (Sigma Aldrich, Milan, Italy).

To assess the relative contribution of extracellular Ca^2+^ entry, the experiments were repeated in the presence of 10 µM BTP-2, an established inhibitor of SOCE [54].

Before starting the experimental protocols, optimal stimulation and muscle length (L_0_) were determined using a series of 1-Hz twitch stimuli while stretching the muscle to a length that generated maximal force (F_0_). After establishing L_0_, muscles were equilibrated using three tetani (500 ms, 150 Hz) administered at 1-min intervals. The setting of muscle parameters was performed at RT (23–25 °C). To evaluate the development of contractures that were induced by high-frequency tetanic stimulation or by external ambient temperature, after waiting 5 min to allow muscle to recover, the muscles were subjected to one of the two protocols described below:

*Ex-vivo exertional stress (ES).* In a first set of experiments, EDL muscles were exposed to an ex-vivo exertional stress protocol, consisting of a series of 150 consecutive tetani (500-ms duration at 80 Hz for each tetanus) applied every 10 s (duty cycle, 0.05), in order to evaluate the development of contractures induced by high-frequency tetanic stimulation. The temperature of the chamber containing the muscles immersed in Ringer’s solution was kept at 30 °C for the entire duration of the protocol (30 min in total). Basal tension was measured every 10 tetani.

*Ex-vivo heat sensitivity (HS).* In a parallel set of experiments, EDL muscles were exposed to an ex-vivo heat stress protocol to assess the development of contractures induced by temperature. Muscles were electrically stimulated with a series of consecutive twitches (1 ms duration at 1 Hz for each twitch) applied every 5 s, while the temperature of the chamber containing the muscles immersed in Ringer’s solution was increased by 2 °C every 5 min from 30 °C to 44 °C. Basal tension was measured at the end of each step of temperature.

At the end of both experimental ex-vivo protocols, muscles were weighted in order to calculate the physiological cross-sectional area (mm^2^) to obtain muscle specific force (mN/mm^2^), as previously described [44].

### 4.4. Preparation of Samples for Light (LM) and Electron Microscopy (EM)

EDL muscles for histological and EM analyses were quickly dissected from the sacrificed control, trained-1 m, and exercised-1 h mice, either before, i.e., in basal conditions, or after the ES protocol. Muscles were pinned on Sylgard dishes and fixed at RT in 3.5% glutaraldehyde in 0.1 M sodium cacodylate (NaCaCO) (Electron microscopy Sciences, Hatfield) buffer (pH 7.4) and stored in the fixative solution at +4 °C until the embedding procedure. Fixed muscles were then post-fixed, embedded, and stained en bloc as previously described [57,58]. Briefly, for histological analysis by LM and standard EM, muscle samples were post-fixed for 1–2 h in 2% OsO_4_. For transverse tubule (TT) staining in EM, specimens were post-fixed in a mixture of 2% OsO_4_ (Electron microscopy Sciences, Hatfield) and 0.8% K_3_Fe(CN)_6_ (Electron microscopy Sciences, Hatfield)for 1–2 h, followed by a rinse with 0.1 M NaCaCO buffer with 75 mM CaCl_2_, and then further processed.

For histological examination, ∼700 nm-thick sections were stained in a solution containing 1% toluidine blue-O and 1% sodium borate (tetra) in distilled water for 3 min on a hot plate at +55–60 °C. After washing and drying, sections were mounted with DPX Mountant for histology (Sigma Aldrich, Milan, Italy) and observed with a Leica DMLB light microscope connected to a Leica DFC450 camera equipped with Leica Application Suite v4.6 for Windows (Leica Microsystem, Wien, Austria).

For standard and TT examination by EM, ultra-thin sections (∼50 nm of thickness) were cut using a Leica Ultracut R microtome (Leica Microsystem, Wien, Austria) with a 45° Diatome Ultra diamond knife (Diatome, Biel, Switzerland) and double-stained with uranyl acetate replacement and lead citrate. Sections were viewed at 60 kV in a FP 505 Morgagni series 268D transmission electron microscope (FEI Company, Brno, Czech Republic), equipped with a Megaview III digital camera (Olympus Soft Imaging Solutions, Munster, Germany) and Soft Imaging System.

## 5. Quantitative Analyses by Electron Microscopy (EM)

For all quantitative EM analyses, micrographs of non-overlapping regions were randomly collected from transverse and longitudinal sections of the internal areas of fast twitch EDL muscle fibers from control, trained-1 m, and exercised-1 h mice, either before, i.e., in basal conditions, or after the ES protocol. The following ultrastructural parameters were evaluated:

*SR-stacks*. The percentage of fibers presenting SR-stacks and the number of SR-stacks per area of section (100 μm^2^) were determined in EM micrographs collected from EDL muscle fibers in transverse sections. In each specimen, 15–20 fibers were analyzed, and, in each fiber, five micrographs were taken at 28,000× magnification.

*Non-triadic TT network at the I band.* We determined both: (i) the extension of the SR in close association with the TT in 100 μm^2^ of section and (ii) the total network of the TT in 100 μm^2^ of section. The non-triadic TT network within the I band was evaluated in micrographs collected from EDL muscle fibers, either stained or not with ferrocyanide, in transverse sections and reported as the average number. In each specimen, 15–20 fibers were analyzed, and, in each fiber, five micrographs were taken at 28,000× magnification.

*Mitochondria.* We determined both the frequency and volume of the mitochondria: (i) The number of mitochondria per area of section (100 μm^2^) was evaluated in micrographs collected from EDL muscle fibers in longitudinal sections and reported as the average number. In each specimen, 10–15 fibers were analyzed, and, in each fiber, five micrographs were taken at 14,000× magnification. (ii) The mitochondrial volume was determined using the well-established stereology point-counting technique [66,67] in micrographs collected from EDL muscle fibers in transverse sections and reported as a percentage of the total fiber volume. Briefly, after superimposing an orthogonal array of dots containing a fixed number of points to the electron micrographs, the ratio between the number of dots falling within mitochondrial profiles and the total number of points covering the whole image was used to calculate the relative volume of fiber occupied by mitochondria. In each specimen, 10–15 fibers were analyzed, and, in each fiber, two micrographs were taken at 8900× magnification.

## 6. Quantitative Analysis by Histology

EDL muscles intended for histological analysis were dissected from control, trained-1 m, and exercised-1 h mice, before (i.e., in basal conditions) as well as after exposure to the ES protocol, prepared for LM (see previous paragraph), and longitudinally sectioned. Images were acquired using a Leica DMLB microscope (Leica Microsystem, Austria) connected to a Leica DFC450 camera (Leica Microsystem, Austria). Histological examination of the EDL muscle fibers was performed as follows: individual fibers were visually scored for rhabdomyolysis based on the presence of extended damage (i.e., contractures, disassembly of contractile elements, extended regions lacking striations). Fibers were classified as: (i) apparently normal and (ii) fibers losing striation (damaged fibers). The number of damaged fibers was counted and reported as a percentage of the total number of analyzed fibers.

## 7. Statistical Analyses

Statistical analysis was determined using Origin 8.0 (OriginLab, Northampton, MA, USA) and Microsoft Excel (Microsoft Office, Bellevue, WA, USA) softwares: The significance was evaluated using Chi, ANOVA, and T-tests, except for time courses of ex-vivo experiments where the significance was evaluated using a repeated-measures ANOVA followed by post-hoc Tukey test for pairwise comparisons. In all cases, data are shown as mean ± SEM, and differences were considered statistically significant at *p* < 0.05.

## 8. Conclusions

The role of external Ca^2+^ in overheating episodes induced by administration of halogenated anesthetics (i.e., MH crises) has been hypothesized [28,29], but its possible contribution to exertional HS has not been directly investigated. Further, the possible contribution of the exercise-induced assembly of CEUs to excessive generation of heat was not considered before. Here, we addressed these issues by promoting the assembly of CEUs with acute exercise and voluntary training and then measured body heat and the generation of muscle tension during ES protocols in-vivo and ex-vivo. Our results support the idea that the entry of external Ca^2+^ may play a significant role in the overgeneration of muscle tension and body heat when exercise is performed in a challenging humid and hot climate.

These findings may lead to the incorrect conclusion that CEUs may be structures detrimental for muscle function. However, we do not believe that this is the case: indeed, we have already extensively demonstrated how CEUs help to limit muscle fatigue and maintain muscle tension during repetitive stimulation (41–46); hence, CEUs are structures that improve muscle function.

Nevertheless, our present results advance the knowledge in the field underlying the following main points: (a) one of the mechanisms that may contribute to exertional HS is excessive entry of external Ca^2+^ through CEUs assembled during exercise; (b) human behavior may often be a factor of risk, as this sometimes involves practicing intensive exercise in hostile climate. For example, large sporting events for such as marathons for amateurs (where many participants pay to compete in a specific event) are planned many months in advance for a specific day and likely will not be postponed (due to organization issues and traveling of participants) when climate conditions are prohibitive. In these cases, many athletes may end up in emergency units due to dehydration and heat-related problems. In the end, animal behavior (which may smarter than human) is more conservative when the climate is adverse; one example is animals living in the desert, which tend to hide from direct exposure to sun rays during the day and postpone many activities to cooler hours at night.

We believe that the results of this study could/should be considered by amateurs, professional athletes, and, especially, by organizers of sporting events to more seriously take climatic conditions into consideration when engaging into strenuous sports activities.

## Figures and Tables

**Figure 1 ijms-23-03772-f001:**
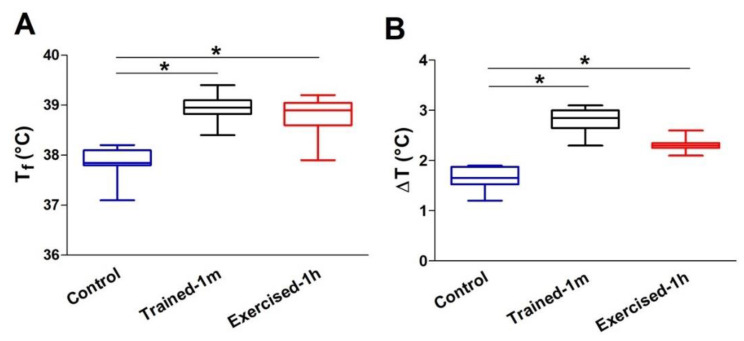
Average core temperature of mice following the ES protocol: (**A**) Average absolute core temperatures recorded at the end of the ES protocol (T_f_); (**B**) average changes in core temperature (ΔT) during ES protocol, i.e., the difference between temperatures measured at the beginning (T_0_) and at the end (T_f_) of the experiment. Data are shown as follows: (i) the box extends from the 25th to 75th percentiles; the line in the middle of the box is plotted at the median, and the whiskers go down to the smallest value and up to the largest (graphs in (**A**) and (**B**); (ii) mean ± SEM (* *p* < 0.05, as evaluated by two-tailed unpaired Student’s *t*-test). Sample size (number of tested mice): control = 8; trained-1 m = 8; exercised-1 h = 9.

**Figure 2 ijms-23-03772-f002:**
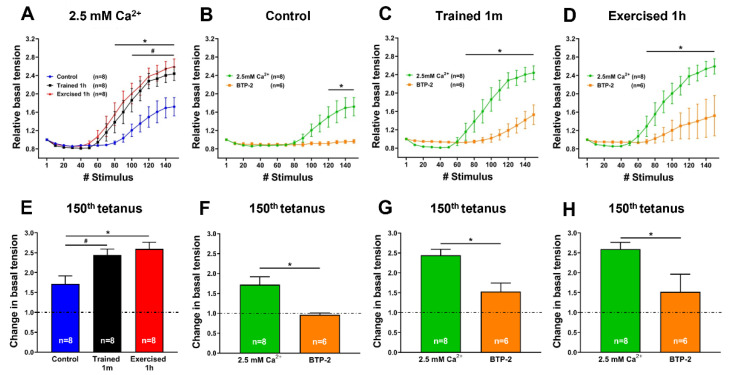
Relative and specific basal tension developed by isolated EDL muscles during an ex-vivo ES protocol: (**A**–**D**) Relative basal tension during an ex-vivo ES protocol in intact EDL muscles either in presence of 2.5 mM of extracellular Ca^2+^ (panel (**A**)) or in a solution supplemented with 10 µM BTP-2 (panels (**B**–**D**)); (**E**–**H**) specific basal tension recorded at 150th tetanus, either in presence of 2.5 mM extracellular Ca^2+^ (panel (**E**)) or in a solution supplemented with 10 µM BTP-2 (panels (**F**–**H**)). Data are shown as mean ± SEM; in panels (**A**,**E**): *^,#^
*p* < 0.05 = * difference between control and exercised-1 h mice; ^#^ difference between control and trained-1 m; in panels (**B**–**D**) and (**F**–**H**): * *p* < 0.05 = difference between 2.5 mM Ca^2+^ and BTP-2; as evaluated by two-tailed unpaired Student’s *t*-test). Note, n = number of EDL muscles analyzed.

**Figure 3 ijms-23-03772-f003:**
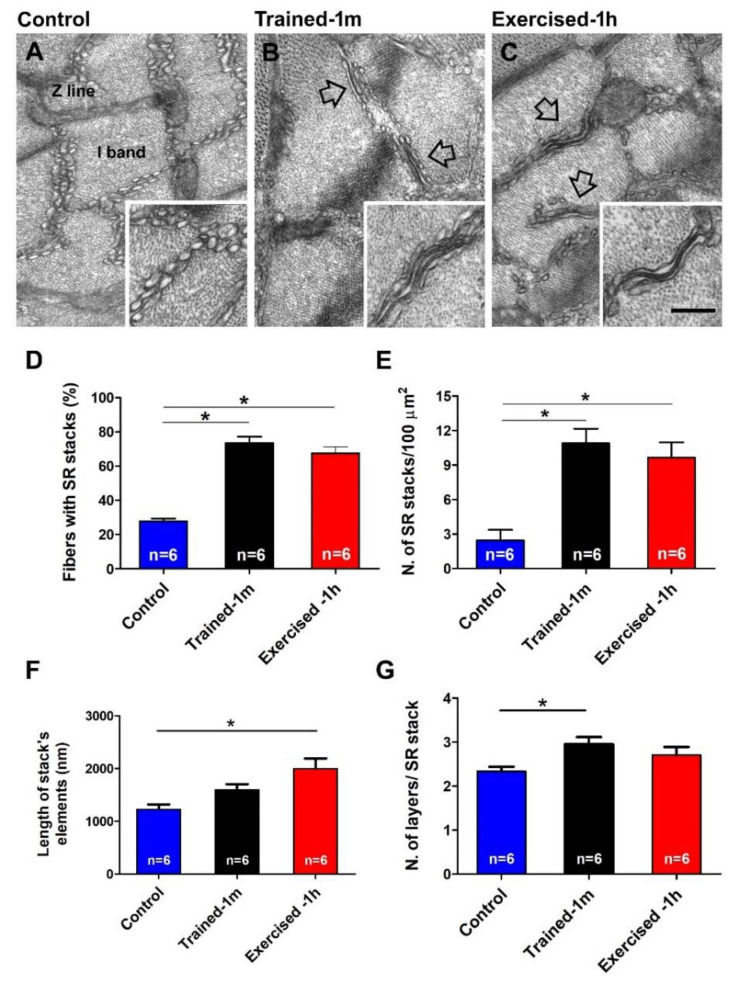
Electron-micrographs and quantitative analysis of SR-stacks: (**A**–**C**) EM images of EDL fibers show the typical organization of the SR at the I band from control (**A**), trained-1 m (**B**) and exercised-1 h (**C**) mice. The inset in (**A**) shows SR vesicles; the arrows and the insets in (**B**,**C**) show the remodeling of SR to form stacks in samples from mice trained-1 m and exercised-1 h. (**D**,**E**) Percentage of fibers containing SR-stacks and number of SR-stacks/100 μm^2^ of section. (**F**,**G**) Length of stack’s elements (nm) and number of layers/SR-stack. Data are shown as mean ± SEM (* *p* < 0.05; the two ends of the line with asterisk indicate which two groups are being compared). Note, n = number of EDL analyzed. Scale bar: (**A**–**C**) = 0.1 μm; insets = 0.2 μm. See also Appendix A for quantitative analysis of SR-stacks after the ES protocol.

**Figure 4 ijms-23-03772-f004:**
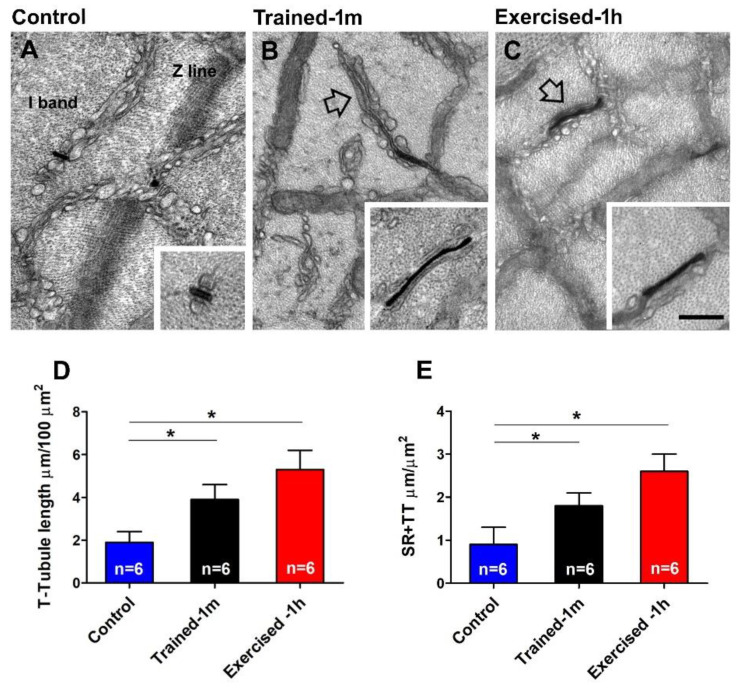
Electron-micrographs and quantitative analysis of the TT network extension at the I band: (**A**–**C**) Representative EM images showing TTs (stained with ferrocyanide-precipitate) at the I band (see arrows and insets in (**B**,**C**)); inset in (**A**) shows a T-tubule stained at the triad junction. (**D**,**E**) Extension of TTs at the I band in 100 μm^2^ of section and extension of SR-TT contacts at the I band. Data are shown as mean ± SEM (* *p* < 0.05; the two ends of the line with asterisk indicate which two groups are being compared). Note, n = number of EDL analyzed. Scale bar: (**A**–**C**) = 0.1 μm; insets = 0.2 μm. See also Appendix A for quantitative analysis of TT extension after the ES protocol.

**Figure 5 ijms-23-03772-f005:**
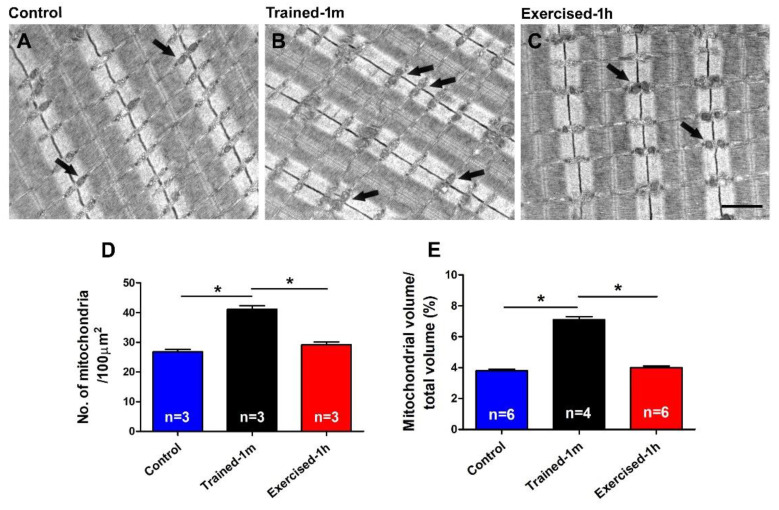
Electron-micrographs and quantitative analysis of mitochondria: (**A**–**C**) Representative EM images showing mitochondria distribution in longitudinal EDL sections (some mitochondria are pointed by arrows); (**D**,**E**) bar graphs showing the quantitative analysis of mitochondria number/area and relative fiber volume occupied by them, respectively. Data are shown as mean ± SEM (* *p* < 0.05; the two ends of the line with asterisk indicate which two groups are being compared). Note, n = number of EDL analyzed. Scale bar: (**A**–**C**) = 1 μm. See also Appendix A.

## Data Availability

Not applicable.

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
