# Peer review of "Store-Operated Ca2+ Entry in Skeletal Muscle Contributes to the Increase in Body Temperature during Exertional Stress"

_ijms, 2022, doi:10.3390/ijms23073772_

Round 1

Reviewer 1 Report

Surprisingly interesting and powerful work devoted to an important problem - the study of the adaptation of skeletal muscles to physical activity in conditions of high temperature and humidity. I was very interested to know about the formation of Calcium Entry Units (CEUs) in adverse environmental conditions and a greater increase in body temperature in mice trained than controls during the ES protocol. An additional Conclusions section reinforces the significance of results for athletes.

I have only minor remarks.

Lane 63

There is no need to decrypt SOCE here, as this has already been done before. Or use the full name. 6 lines of Frequently used Abbreviations - don't you think it's too much, and complicating the understanding of the text? Perhaps some of the abbreviations that are not common in use can be written in an unabbreviated form, which will make the text more accessible to an unprepared reader.

Is it possible to highlight the novelty of the work, were such data or their part known before?

Figures

It didn’t immediately become clear to me personally what the asterisks and the lines below them on the figures. If this indicates a significant change relative to control, then it may be sufficient to mark the column itself with an asterisk. Perhaps this is a generally accepted designation and it seemed not obvious to me.

Please correct some typos, for example, on lane 14 (1_month), lane 15, (1_hour), lane 305 (for_1_hour).

I would not dot the titles (line 91 onwards).

Reference 52 - please add the journal and page number.

Author Response

Answers to Reviewers.

Reviewer n. 1

Surprisingly interesting and powerful work devoted to an important problem - the study of the adaptation of skeletal muscles to physical activity in conditions of high temperature and humidity. I was very interested to know about the formation of Calcium Entry Units (CEUs) in adverse environmental conditions and a greater increase in body temperature in mice trained than controls during the ES protocol.

An additional Conclusions section reinforces the significance of results for athletes.

We thank the Reviewer for his/her interest in the present work and for the comments that helped us to improve the quality of the manuscript.

I have only minor remarks.

Comment 1. Lane 63

There is no need to decrypt SOCE here, as this has already been done before. Or use the full name.

Corrected

Comment 2. 6 lines of Frequently used Abbreviations - don't you think it's too much, and complicating the understanding of the text? Perhaps some of the abbreviations that are not common in use can be written in an unabbreviated form, which will make the text more accessible to an unprepared reader.

We have now significantly reduced the number of abbreviations listed, deleting those that were indeed not frequently used.

Comment 2. Is it possible to highlight the novelty of the work, were such data or their part known before?

We thank the reviewer for having asked to do this. Thank you. Yes, these data are novel: for the first time we tested if exercise-mediated assembly of Ca2+ entry units (first discovered by us in 2017), hence increased number of sites that allows Ca2+ entry, could determine hyperthermia.

We have now highlighted the novelty of these data as follow:

In Introduction:

Line 84: However, whether SOCE contributes to exertional HS still remains to be determined.

Line 112: To determine whether SOCE may play a role in exertional HS [28-30], in the present work we tested the hypothesis that Ca2+ entry via CEUs may increase the risk of hyperthermic crisis when physical activity is performed in challenging environmental conditions. The results collected in this study show for the first time how exercise-induced assembly of CEUs results in: a) hypersensitivity of muscle to develop contracture; and b) excessive increase in internal temperature of mice subjected to exertional stress (ES).

In Results:

Line 228: Note that, while assembly of SR-stacks during acute exercise was already reported [44,49], the assembly of SR-stacks following voluntary training in wheel cages is novel, i.e. not previously reported (Fig. 3 B).

In Discussion:

Line 319: However, whether SOCE contributes to exertional HS has never been addressed before.

Line 321: As we have recently discovered that exercise induces remodelling of SR and TT (to form CEUs), increased STIM1/Orai1 aggregation, enhanced Ca2+ entry via SOCE, and finally improved fatigue resistance [44; 47-50], in the present work we tested for the first time the hypothesis that exercised-induced assembly of CEUs may contribute to excessive generation of heat in mice subjected to ES in hot/humid environmental conditions. To accomplish this, we compared mice and isolated muscles from control animals and from animals with pre-assembled CEUs (a. trained for 1 month in wheel cages or b. excised 1 hour with a protocol that we know induces CEU assembly). 

Line 351: The assembly of CEUs is also promoted by voluntary training in wheel cages: this result is novel, i.e. it was not previously reported (Figs. 3 and 4).

In Conclusions:

Line 566: The role of external Ca2+ in overheating episodes induced by administration of halogenated anesthetics (i.e. MH crises) has been hypothesized [28,29], but its possible contribution to exertional HS has not been directly investigated. Also, the possible contribution of exercise-induced assembly of CEUs to excessive generation of heat was not considered before. Here we addressed these issues by…..

Line 594: We believe that the results of this study could/should be considered by amateurs, professional athletes, and especially by organizers of sport event to take more seriously into consideration the climatic conditions when engaging into strenuous sport activities.

Figures

It didn’t immediately become clear to me personally what the asterisks and the lines below them on the figures. If this indicates a significant change relative to control, then it may be sufficient to mark the column itself with an asterisk. Perhaps this is a generally accepted designation and it seemed not obvious to me.

We have used this graphical appearance of statistical differences many times in the past 5 years, in diverse journals, and it generally accepted. Please see the following publications: 

Boncompagni S., et al. 2017. Scientific Reports. 10.1038/s41598-017-14134-0.

Michelucci A., et al. 2020. J Gen Physiol. 10.1085.

Pecorai C., et al. 2020. Front Physiol. 10.3389.

I will try to explain showing the example of Figure 2:

In Figure 2 A, the lines-with-asterisks (or the hashtag) indicates a range in which data are significantly different between two specific groups.  We have updated the legend to make things clearer:

In panels A: *#p<0.05 = *difference between control and exercised-1h mice; #difference between control and trained-1m.

In panels C-D: lines-with-asterisks indicates the range in which *difference between 2.5mM Ca2+ and BTP-2 is significant.

On the other hand, when the line-with-asterisk in above two bars (example: the bar graphs in panels E), the two ends of the line indicate which two groups are being compared. For example: in Fig 2 E the top-line compares control and exercised 1h, while the bottom line compares control with trained 1m.

In panels E: *#p<0.05 = *difference between control and exercised-1h mice; #difference between control and trained-1m.

In panels F-H: *p<0.05 = *difference between 2.5mM Ca2+ and BTP-2.

In the legends of Figures 3, 4 and 5 we have implemented the legend as follow:

(*p < 0.05; the two ends of the line-with-asterisk indicate which two groups are being compared).

Please correct some typos, for example, on lane 14 (1_month), lane 15, (1_hour), lane 305 (for_1_hour).

Corrected

I would not dot the titles (line 91 onwards).

Done

Reference 52 - please add the journal and page number.

Done: reference 52 was published in J Muscle Res Cell Motil. 42:233–249.

Reviewer 2 Report

Summary

This paper entitled “store-operated Ca2+ entry in skeletal muscle contributes to the increase in body temperature during exertional stress” describes a comprehensive study about the role of extracellular calcium in formation of calcium entry units at high temperature and their potential effect on muscle function in hyperthermic condition which often leads to heat stroke in animals. The authors employed a set of sophisticated methods including physiology and EM to study whether physical exercise could increase the risk of hyperthermic events via calcium entry units. They found that pre-exercised and trained mice exhibited higher body temperature when faced to challenging heat environments. Ex-vivo physiology experiments revealed EDL muscles isolated from the pre-exercised mice generated a larger basal tension, and they contained more calcium entry units.

Overall, the study represents sophisticated and impressive technical approach. This research is carefully done and well-described, representing an important contribution to our understanding of how physical activity shapes our body in response to challenging environment. I only have a few comments to improve this paper prior to publication.

Comments

  1. Although the authors showed that pre-exercised and trained mice exhibited higher body temperature after ES, whether the anatomical and physiological differences observed in these mice was due to the increased body temperature is not clear. Is there any potential link between increased body temperature and calcium changes in muscle cells, or these may be irrelevant?
  2. Figure 1. Data in the bottom table have been presented in panel A and B. Either delete the table or move it to supplemental tables.
  3. Figure 3. In Panel A, where the inset picture highlight in the main image? Also there are more than one arrows in panel B and C, which one was highlighted in the inset images?
  4. Figure 5. Panel D is confusing: what does the y-axis stand for? It will be much clearer if the panel D could split into two bar-graphs. One for mitochondria numbers and the other one for volume.

Author Response

Answers to Reviewers.

Reviewer n. 2

Summary

This paper entitled “store-operated Ca2+ entry in skeletal muscle contributes to the increase in body temperature during exertional stress” describes a comprehensive study about the role of extracellular calcium in formation of calcium entry units at high temperature and their potential effect on muscle function in hyperthermic condition which often leads to heat stroke in animals. The authors employed a set of sophisticated methods including physiology and EM to study whether physical exercise could increase the risk of hyperthermic events via calcium entry units. They found that pre-exercised and trained mice exhibited higher body temperature when faced to challenging heat environments. Ex-vivo physiology experiments revealed EDL muscles isolated from the pre-exercised mice generated a larger basal tension, and they contained more calcium entry units.

Overall, the study represents sophisticated and impressive technical approach. This research is carefully done and well-described, representing an important contribution to our understanding of how physical activity shapes our body in response to challenging environment. I only have a few comments to improve this paper prior to publication.

We thank the reviewer for the appreciation of our work.

Comments

Although the authors showed that pre-exercised and trained mice exhibited higher body temperature after ES, whether the anatomical and physiological differences observed in these mice was due to the increased body temperature is not clear. Is there any potential link between increased body temperature and calcium changes in muscle cells, or these may be irrelevant?

I am not sure if I understood properly the question, but I will try my best to respond appropriately to this query.

We have previously shown that a fatigue protocol on treadmill (approximately 1 hour of exercise) induced remodeling of the SR and of TT leading to formation of CEUs (Boncompagni et al. 2017 and 2018; Michelucci et al. 2019; refs n. 44, 48, 49 of the present paper). In those papers, mice were subjected to the fatigue protocol at room temperature.

In the present paper we used exactly the same protocol at room temperature to induce assembly of CEUs in the exercised-1hour group. Also trained-1month mice exercised in wheel cages at room temperature.

The quantitative analysis of CEU-assembly in Figs. 3 and 4 was done before the exertional stress (ES) protocol (1hr at 34°C and 40% humidity) was applied: these data indicates that muscle fibers of both exercised-1hour and trained-1m mice contains a greater number of CEUs than controls. Hence, we must conclude that the remodeling of SR and TT that leads to formations of CEUs is not the result of exposure to heat, but the consequence of exercise performed at room temperature.

So, the direct answer to the query would be:

“No, the anatomical and physiological differences observed in these mice (trained/exercised vs. control) are not due to the increased body temperature, but to the presence in muscle of pre-assembled CEUs before the exposure of mice to the ES protocol”.

Figure 1. Data in the bottom table have been presented in panel A and B. Either delete the table or move it to supplemental tables.

We have now moved data in Figure 1 to the Supplemental Materials file (see Suppl. Table 1).

Figure 3. In Panel A, where the inset picture highlight in the main image? Also there are more than one arrows in panel B and C, which one was highlighted in the inset images?

Only the inset in panel C was taken from the main image (but rotated). For consistency, none of the 3 insets of Figure 3 are now enlarged details of the 3 main panels, but images at higher magnification of the SR taken from a different area of the same sample. The insets show the morphology of the SR at the I band in control samples (panel A) and the remodeling induced by exercise, i.e. the SR stacks (in panels B and C).  Empty arrows in panels B and C of Figure 3 point to SR-stacks, as indicated in the figure legend.

Figure 5. Panel D is confusing: what does the y-axis stand for? It will be much clearer if the panel D could split into two bar-graphs. One for mitochondria numbers and the other one for volume.

We were measuring parameters that were labeled at the bottom. However, for clarity, we have now splitted panel D of Figure 5 in two separate bar-graphs, obtaining a 5 panels Figure.  The y-axis of panel D is now the number of mitochondria / 100 mm2, while the y-axis of panel E is the relative fiber volume occupied by mitochondria (expressed as a percentage of total).